# Visiting Molecular Mimicry Once More: Pathogenicity, Virulence, and Autoimmunity

**DOI:** 10.3390/microorganisms11061472

**Published:** 2023-06-01

**Authors:** Yuri Chaves Martins, Arnon Dias Jurberg, Cláudio Tadeu Daniel-Ribeiro

**Affiliations:** 1Department of Anesthesiology, Saint Louis University School of Medicine, St. Louis, MO 63110, USA; 2Instituto de Educação Médica, Campus Vista Carioca, Universidade Estácio de Sá, Rio de Janeiro 20071-004, RJ, Brazil; 3Laboratório de Animais Transgênicos, Universidade Federal do Rio de Janeiro, Rio de Janeiro 21941-599, RJ, Brazil; 4Laboratório de Pesquisa em Malária and Centro de Pesquisa, Diagnóstico e Treinamento em Malária, Instituto Oswaldo Cruz, Fiocruz, Rio de Janeiro 21041-250, RJ, Brazil

**Keywords:** antigen sharing, autoimmunity, bioinformatics, host/parasite relationship, immune tolerance, parasite

## Abstract

The concept of molecular mimicry describes situations in which antigen sharing between parasites and hosts could benefit pathogen evasion from host immune responses. However, antigen sharing can generate host responses to parasite-derived self-like peptides, triggering autoimmunity. Since its conception, molecular mimicry and the consequent potential cross-reactivity following infections have been repeatedly described in humans, raising increasing interest among immunologists. Here, we reviewed this concept focusing on the challenge of maintaining host immune tolerance to self-components in parasitic diseases. We focused on the studies that used genomics and bioinformatics to estimate the extent of antigen sharing between proteomes of different organisms. In addition, we comparatively analyzed human and murine proteomes for peptide sharing with proteomes of pathogenic and non-pathogenic organisms. We conclude that, although the amount of antigenic sharing between hosts and both pathogenic and non-pathogenic parasites and bacteria is massive, the degree of this antigen sharing is not related to pathogenicity or virulence. In addition, because the development of autoimmunity in response to infections by microorganisms endowed with cross-reacting antigens is rare, we conclude that molecular mimicry by itself is not a sufficient factor to disrupt intact self-tolerance mechanisms.

## 1. The Adaptive Immune Response and the Maintenance of Tolerance to Self

One of the main paradigms of immunology is the notion that the immune system has mechanisms that halt the formation of autoimmune responses. Rather than representing an absence of response, immunological tolerance is an active process that occurs in both primary and secondary lymphoid organs [1,2,3]. When tolerance mechanisms are functioning satisfactorily, we generate immune responses to practically all microorganisms and foreign antigens in the body, while avoiding harmful immune responses to our own components (self).

However, occasionally, the mechanisms of tolerance to self-antigens are breached, and immune reactions against self-antigens can arise. Autoimmune reactions involve both the production of autoantibodies and the generation of autoreactive cells that can damage tissues in autoimmune diseases [2,3,4]. Although they have steadily risen throughout westernized societies, it is estimated that the prevalence of autoimmune diseases is limited to approximately 3–10% of the world population [5].

All organisms that inhabit the planet are made up of a finite set of redundant components that were conserved during evolution. Of paramount importance, these building blocks can share great similarity between species that are evolutionarily very distant [6], indicating that some structures may be essential for species survival. Moreover, convergent and parallel evolution allows the development of similar structures by independent species that share common environments [7]. Thus, one of the proposed mechanisms for the disruption of immunological tolerance is the exposure of the organism to external antigens that are too similar to self-antigens from a structural or conformational point of view. In this case, the host develops an immune response to antigens shared upon infection, at the risk of developing an autoimmune response and autoimmune pathology (Figure 1).

A number of pathogens exhibiting structures similar to human proteins has been reported [2]. For example, some antibodies produced against *Streptococcus pyogenes* protein M type 5, during oropharynx infection with β hemolytic group A *S. pyogenes,* can also react to antigens present in the cardiac valves (cardiac myosin), causing rheumatic endocarditis [8]. Other autoimmune diseases which generate results from the cross-reactivity between microbial and self-antigens include ankylosing spondylitis [9], Chagas cardiomyopathy [10]; type 1 diabetes mellitus [11], multiple sclerosis [12], Guillain–Barré syndrome [13], chronic inflammatory demyelinating polyneuropathy [14], Lyme disease arthritis [15], autoimmune polyglandular syndrome [16], primary biliary cirrhosis [17] and tropical spastic paraparesis [18].

However, the production of pathogenic autoantibodies resulting from the cross-reactivity of antigens between parasites and their hosts is far from usual, corresponding rather to exceptions to the rule. In most cases, only non-pathogenic autoantibodies are produced during infections, such as the production of autoantibodies against cardiolipin complexed with lecithin and cholesterol (VDRL) during *Treponema* spp. infections [19] or the formation of autoantibodies during malaria [20].

A population of autoantibodies, known as “natural autoantibodies”, is permanently produced, even before the contact of organisms with any external antigen. The function of these autoantibodies is not clearly established yet, although the increased ability of these molecules to cross-react with external antigens suggests that they function as a first barrier in the anti-microbial response [21]. Similarly, the physiology of maturing lymphocytes during differentiation includes a mechanism (called positive selection) in which the T lymphocytes that react weakly to autoantigens in the thymus will integrate the repertoire of mature and differentiated cells [22]. A tonic signal given by endogenous peptide/MHC complexes also favors the survival of mature T cell in the periphery [22]. Thus, autoimmune disease rarely accompanies most autoimmune responses, including those generated through exposure to microbial antigens [2,20].

## 2. Molecular Mimicry and Its Implication as a Mechanism of Parasite Escape from the Host Immune Response

Considering that the maintenance of tolerance to self-antigens is the rule, Raymond Damian coined the concept of “molecular mimicry” in 1964, based on the observation of antigen sharing between parasites and hosts [23]. According to him, parasites would tend to develop structures similar to those of their hosts, with the consequence that they are not recognized as strange to the infected organism. The escape from the host immune response happens because hosts have central and peripheral tolerance mechanisms such as negative B and T cell selection, regulatory T cells, and anergy (for review see [2]). Hence, infectious organisms carrying antigens common to their hosts would more easily escape from the immune response, having a selective advantage. These mimicry molecules can be perfect mimics when they co-opt host factors or imperfect when they resemble host components and yet perform distinct functions which confer an advantage to the pathogen [24]. Damian based his idea on the concept of mimicry that was created in the 19th century by Henry Walter Bates to explain why certain butterflies in the Brazilian Amazon, considered tasty by their predators, resembled other less palatable species, thereby avoiding predation [25].

The first evidence of molecular mimicry was obtained using immunoprecipitation methods, showing that anti-serum against parasites also reacted to host antigens [26,27,28]. Subsequently, new technologies such as monoclonal antibodies, molecular cloning and proteome comparison were used to confirm that parasites and other infectious organisms typically share antigens with their hosts [29,30,31,32,33,34,35,36,37,38,39,40]. Later, the concept of molecular mimicry was expanded, and four different types of mimicry have been described: (1) similarity in the sequences and structures of full-length proteins or domains, (2) structural similarity without sequence homology, (3) similarity in protein short linear motifs, also known as motif mimicry, and (4) similarity of binding surface architectures even without sequence homology, known as interface mimicry [7].

While pathogenic and non-pathogenic organisms use molecular mimicry to escape the host immune response and other barriers imposed to their survival in the infected organism, the hosts also develop and acquire mechanisms that hinder or impede the establishment of infections. One such escape mechanism is known as “reverse mimicry”, which occurs when hosts acquire or develop antigens or other molecules that are similar to those present in pathogens and that facilitate resistance or tolerance to infections. For example, sheep use endogenous retrovirus proteins as restriction factors that block pathogenic retrovirus infections [41].

In this interaction, which occurs during the co-evolution of pathogens and their hosts, one could consider that each organism is trying to defeat the defenses imposed by the other. However, the result of this permanent “arms race” would be the permanence of both organisms in the same place of the host parasite relationship. The evolutionary biologist Leigh Van Valen originally proposed this concept and called it “the red queen hypothesis” [42]. The name derives from a passage in the book by Lewis Carroll (1832–1898) *Through the Looking-Glass, and What Alice Found There* in which the Red Queen tells Alice: “[I]t takes all the running you can do, to keep in the same place” [43].

Apart from helping parasites and hosts to remain in the same place, molecular and reverse mimicry can also increase parasite–host antigenic similarity over time. Should this hypothesis be true, two parasites phylogenetically equally distant from their hosts would significantly differ in the proportion of antigen similarity if the history of host–parasite association greatly varies between pairs. More specifically, it would be expected to find higher antigen similarity in longer-lasting evolutionary interactions than in more recent ones.

## 3. Protein Identity Screening by Computational Analysis

To date, discovery of molecular mimicry examples has been made largely on a case-by-case basis, and it is possible that there exist many additional mimetic proteins that may be detectable through computational methods. Studies comparing the genomes and proteomes of different species are now possible using bioinformatics and genetic sequencing approaches. Because the degree of protein similarity increases as the evolutionary distance between species reduces, the degree of protein identity can be used to infer phylogeny [6,44,45,46]. In addition, these methodologies enable studies on the relationships between infectious agents and their hosts. The antigenic similarities between the hosts and infectious microorganisms may be useful for understanding the success or failure of the parasitism between these organisms [47]. This understanding can also reveal potential implications of antigen sharing and molecular mimicry with respect to the maintenance of tolerance and the prevention of autoimmunity [2].

Studies comparing the proteomes of infectious agents and their hosts with the goal of screening for mimicry proteins have been done before [36,38,47,48,49,50,51,52,53]. An analysis of published studies suggests a lack of criteria for screening protein mimicry. For example, some studies have excluded proteins that show similarity between the genomes of the parasite and its host if these molecules are also present in phylogenetically close non-pathogenic organisms or organisms in an intermediate position in the evolutionary scale between the parasite and the host [50]. This approach produces reduced numbers of false positive hits, but consequently increases the number of false negatives as relevant motifs/proteins for host immune evasion may be also present in non-pathogenic or in phylogenetically unrelated species. Other studies used only peptide patterns to compare proteomes and found extensive peptide overlap between viruses and humans or bacteria and humans [36,47,48,52,54]. A drawback of this strategy, however, is the high number of false positive hits that can mislead further investigations. This absence of criteria can lead to different interpretations of the same data set.

Amino acid sequence-based approaches can also return false positives by excluding conformational similarities. This arises when similar amino acid sequences occupy different positions in protein pairs. Whereas one of them may be on the protein surface, the other motif may assume a cryptic position upon protein folding. Likewise, sequence analysis does not consider post-translational modifications, which may create novel immunogenic epitopes.

To add new insights to this discussion, we estimated the similarity between human proteins as a reference and organisms with varying phylogenetic distances to humans using the proteome comparison tool of the Bacterial and Viral Bioinformatics Resource Center (BV-BRC) [55]. In total, we retrieved the proteomes of 46 organisms from the Uniprot database for comparison (Appendix A). The species were chosen to have at least one representative of all three domains of life. An all-by-all BLAST analysis was performed, in such a way that all human proteins were compared with all bacterial proteomes (for a detailed description of the methods, please see [56]). Because of the dispute over the criteria to define protein mimicry [47,48,50,52], we evaluated both low (sequence coverage greater than 30% and identity greater than 10%) and high (sequence coverage greater than 70% and identity greater than 70%) stringency parameters. As another significance threshold for protein similarity, we evaluated three different expect values (e-values) as a parameter of peptide sequence overlap between different protein pairs. Thus, high e-values (i.e., 1^−10^) were obtained from the occurrence of alignments by chance, which are more likely to occur with shorter amino acid sequences or if protein pairs share a few motifs. In turn, the reduction of the e-value (i.e., 1^−100^) indicates proteins that are practically identical.

As anticipated, the number of proteins with at least one similar human protein was higher when less stringent parameters were used, and a sharp decrease occurred when more stringent parameters were considered. For example, the number of similar proteins per species when considered an e-value of 10^−50^ showed a median value of 1157 (95% CI: 389–1770) with low stringency parameters and a median value of 56 (95% CI: 1–98) with high stringency parameters (Figure 2A). The number of similar protein pairs also significantly decreased as e-values were lowered when low stringency parameters were evaluated (Figure 2B). However, decreases in the e-value did not significantly change the number of similar protein pairs identified when high stringency parameters were considered (Figure 2C). In addition, as expected, we found that species phylogenetic close to humans, such as rats and mice, exhibited the highest numbers of similar proteins to the reference. Together, our observations suggest that changing the value of a single parameter was sufficient to produce a significant difference in the number of proteins considered similar for all the organisms studied. These data show that the sensitivity and specificity of proteome comparison studies using bidirectional BLASTP for screening of potential mimetic proteins can vary substantially after changing just a single criterion. Therefore, further studies are needed to optimize the sensitivity and specificity of this methodology for the identification of proteins with a possible mimetic function.

However, detection of sequence similarity between host and pathogenic proteins is by itself not indicative of mimicry or pathogen-specific exploitation of host functions. Hence, proteome comparison studies are just the initial step to identifying possible mimetic proteins and can be used only as a screening test. These studies should be followed by a dedicated domain-centric sequence and structural analysis to examine the potential functions of the screened proteins and pathogenesis mechanisms [7,57]. This has been done in the past on a case-by-case basis with different pathogenic organisms [2,7,8,9,11,12,13,15,17,29,30,35,49,54,58,59,60].

More likely, there is no perfect approach to identifying potential mimetic proteins in all cases and different sets of parameters can be used in a complementary way according to the characteristics of the proteomes being compared and the objectives of the comparison.

Computational analysis studies can then guide directed in vivo experimental studies to confirm the role of a screened protein in the avoidance of the host immune response. One possibility to functionally evaluate mimicry candidates is to immunize animals with different peptide versions harboring specific features, such as the deletion or replacement of any given motif or the substitution of residues that are post-translationally modified. Another strategy would be to take advantage of CRISPR methods to carry out small deletions or substitutions in the parasite genome. These genetically modified parasites can then be used to infect wild-type experimental hosts and the immune response can be further assessed. It would therefore be expected that alterations in relevant mimicry sites would modify parasite virulence.

## 4. Antigen Sharing and Host–Parasite Co-Evolution

Despite the limitations of bioinformatic tools to identify true protein mimicry, would it be possible to estimate the average number of antigens that we share with a microorganism using computational methods? The first study that examined the fully sequenced human genome revealed that less than 1% of our genes are species-specific [61]. However, innate and humoral immune responses are not based on the recognition of whole molecules, but of small molecular motifs such as pathogen-associated molecular patterns (PAMPs) or antigens. For example, the optimal length of a linear B cell epitope is five amino acids [62], although epitopes of up to 16 amino acids in length have been reported [63,64]. In addition, there is evidence of an extensive T-cell epitope repertoire shared among the human proteome and our gut microbiome [54,65]. Therefore, a protein does not necessarily need near-complete identity to have a mimicry function.

Should the human proteome be divided into short pentapeptides (5-mers) and compared to same-sized peptides derived from pathogenic or non-pathogenic viruses and bacteria, more than 89% of them exhibited perfect identity [48,52]. However, a sharp decrease in the percentage of similar peptides was observed when longer oligopeptides were considered. More specifically, peptide similarities dropped to 28.8–37.5% when hexapeptides were considered, 3.0–4.9% for heptapeptides and 0.4–0.7% for octapeptides [52]. The authors of these studies argue that this level of redundancy is not stochastic, but rather reflects the strong evolutionary persistence of certain peptides that would be vital for the function of proteins in viral, bacterial and human proteomes [48,52]. In addition, certain pentapeptide and hexapeptide combinations are absent from all publicly available proteome sequences [66], reinforcing this hypothesis. The fact that viruses and bacteria exhibit 89% of 5-mer peptide similarity with the human proteome suggests that the human immune system is continuously exposed to a large number of self-like antigens produced by parasites and mutualists. Nevertheless, this methodology does not indicate an average number of proteins with similar motifs between two organisms considering all possible protein epitopes of different sizes. In addition, it excludes molecular mimicry arising from the acquisition of host cell genes through horizontal gene transfer, which can be evidenced by sequence similarity with host proteins and by phylogenetic analyses.

We decided to calculate the percentage of proteins with similar motifs between humans and selected common pathogenic and nonpathogenic organisms using the BV-BRC proteome comparison tool [55] by carrying out a subset analysis of our previous dataset. In total, we included the proteomes of 25 pathogenic and 13 nonpathogenic organisms (Appendix A). We again considered both low and high stringency parameters and three different e-values (Figure 3).

The percentage of similar proteins per species when we considered low stringency parameters showed a median value of 38.73% (95% CI: 33.54–43.22%) with an e-value of 1^−10^ that significantly decreased to a median value of 3.32% (95% CI: 2.59–5.61%) with an e-value of 1^−100^ (Figure 3A). A sharp decrease in the percentage of homologous proteins was also observed when high stringency parameters were considered with a median value of 0.11% (95% CI: 0.02–0.9%) with an e-value of 1^−10^ and a median value of 0.05% (95% CI: 0–0.56%) with an e-value of 1^−100^ (Figure 3B).

As we have considered homologies between proteins derived from diverse proteomes independently of their sizes, we have obtained relatively smaller numbers than those generated from analyses comparing only pentapeptides. However, these results are still impressive, as more than one-third of the proteins of some organisms showed protein sequence similarities to human proteins when considering low stringency parameters. Our data are also in agreement with Doxey and McConkey who analyzed the proteomes of 128 bacterial species and found that 27.4% of the human proteins had BLAST matches in one or more bacterial proteomes considering an e-value of 10^−6^ [57]. Therefore, our data corroborate the assumption that antigen sharing between humans and common pathogenic and non-pathogenic organisms is substantial. The questions that naturally arise from this conclusion are: (1) Is the degree of antigen sharing related to pathogenicity? (2) What is the risk of antigen sharing for the development of autoimmunity?

## 5. Antigen Sharing and Its Relationship with Pathogenicity and Virulence

There is little consensus in the literature about the definitions of pathogenicity and virulence with both terms frequently used interchangeably [67,68]. For this work, we used the following definitions: (1) Pathogenicity is the ability of an organism to infect a host and cause disease; (2) Virulence is the relative capacity of a microbe to cause direct or indirect damage and/or disease to a host, with the term “relative” being a necessary component of the definition because there are no absolute measures of virulence. Virulence is nearly always obtained from hosts that are already infected. Therefore, pathogenicity can be used as a qualitative term meaning that it is an “all-or-none” concept (the microbe is either pathogenic or not), whereas virulence is quantitative (a parasite can have different degrees of virulence depending on its hosts). In addition, pathogenicity is a broad qualitative term that encompasses host–pathogen interactions with different degrees of virulence.

Molecular mimicry, being a strategy used by pathogens to evade host defenses, plays a role in a wide range of pathogenic factors such as virulence pathways, evasion of host immune response, intracellular survival in host cells and the development of autoimmune reactions [2,7,53,57]. This suggests that there may be a relationship between the degree of antigen sharing and pathogenicity.

Motivated by the broad goal of detecting a relationship between host–pathogen antigen sharing on a genomic scale and pathogenicity, we compared the percentage of similar proteins to the human proteome between 25 pathogenic vs. 13 non-pathogenic organisms (for a full list of the organisms, please see Appendix A). Based on the results shown in Figure 2, we considered only low stringency parameters and three different e-values (Figure 4A).

The percentages of similar proteins were not significantly different between pathogenic and non-pathogenic organisms in all tested e-values (Figure 4A). Non-pathogenic organisms showed a median value of 35.74% (95% CI: 28.69–40.72%) with an e-value of 1^−10^ compared to a median value of 41.83% (95% CI: 33.98–46.91%) for pathogenic organisms. As expected, the percentages of similar proteins significantly decreased in both groups when smaller e-values were considered with a median value of 3.2% (95% CI: 2.06–8.34%) for non-pathogenic organisms and 3.7% (95% CI: 2.62–5.90%) with an e-value of 1^−100^ (Figure 4A). In conclusion, pathogenic organisms did not present a significant increase in the percentage of antigen sharing with the human proteome when compared to non-pathogenic organisms in our sample.

Although antigen sharing was not associated with pathogenicity, because molecular mimicry, per definition, can enable immune evasion, we hypothesized that the degree of antigen sharing in a host–pathogen interaction could be associated with virulence. The most used measurement of virulence is the lethal dose required to kill 50% of infected hosts, referred to as the LD50. However, we wanted a measure that not only would allow comparisons across pathogens, but also would be applicable to host–pathogen systems where host death does not occur by accounting for other outcomes of infection such as chronicity and latency. Casadevall [69] proposed the concept of pathogenic potential that is a refined measure of virulence of a host–pathogen system and calculated it for common mice pathogens. We tested for a possible correlation between the pathogenic potential of 24 species (Appendix A) and their percentage of proteins with similar motifs to the mice proteome. We again considered both low and high stringency parameters and three different e-values.

The percentages of similar proteins were not correlated with the pathogenic potential of the species in all conditions analyzed (Figure 4B,C). Spearman correlation coefficients varied between −0.01 (95% CI: −0.42 to 0.40) and 0.27 (95% CI: −0.15 to 0.61) and p-values were always greater than 0.05.

In summary, our data indicate that the degree of protein sequency similarity in host–pathogen systems is not related to pathogenicity and virulence. This may be because molecular mimicry is also involved in other types of interactions between organisms such as commensalism, mutualism and amensalism [7,16,24,54,57]. For example, selection pressure to evolve mimetic molecules that aid in colonizing the host would be similar in both commensal non-pathogenic bacteria as well as pathogens [57]. Further, our approach did not consider molecular mimicry derived from the structural similarity without sequence homology which could be the type of mimicry that predominates in the pathogenic and non-pathogenic organisms analyzed.

## 6. The Risk of Antigen Sharing to the Development of Autoimmunity

Sustained autoimmune manifestations are considered rare events in a population [2,70,71]. The total number of antigens that are potential targets for autoimmune responses (estimated through the known number of autoantibody specificities) is approximately in the low hundreds [72]. This means that less than 1% of the approximately 30,000 human proteins would be involved in autoimmunity. The number of known autoimmune diseases is small. Roitt’s Essential Immunology [70] index includes 31 major diseases in the chapter on autoimmune diseases in this celebrated and classic immunology textbook, 10 (or fewer) of which correspond to the most common and well-known diseases. A review of the scientific literature identified 81 autoimmune diseases [71]. In addition, as far as we know, a total of 23 types of autoimmune diseases is associated with molecular similarity between humans and mutualist/parasite organisms [36]. These estimates are not consistent with the highest potential number of antigens shared between humans and pathogenic/non-pathogenic organisms (approximately 90%—27,000 of our constituent proteins considering the homology of pentapeptides, or approximately 3.3%—990 proteins—using an e-value of 10*^−^*^100^), which could potentially induce autoimmunity.

Two non-mutually exclusive rationales could explain this contrast: (a) the existence of extremely efficient mechanisms of tolerance that would fail only in rare cases; (b) the phenomenon of antigen-sharing would not be a predominant mechanism of breaking immunological tolerance, with the consequence that the development of autoimmunity and/or autoimmune pathology would not occur most of the time.

In fact, the mechanisms involved in the maintenance of tolerance to autoantigens are multiple and redundant, making the context in which these autoantigens are presented to the immune system only one factor predisposing the development of autoimmunity. For instance, the balance between autoimmunity and self-tolerance is affected by factors inherent to the hosts such as the development of auto-reactive T cells in response to cryptic antigens or environmental chemicals [3,12,22,51].

Therefore, it can be assumed that jawed vertebrates contact thousands of homologous (auto) antigens displayed in “parasitic clothes” daily and that this event is not sufficient to determine the genesis of autoimmune pathology, as the production of pathogenic autoantibodies through exposure to microorganisms with cross-reactive antigens is minimal in the presence of effective regulatory mechanisms for an enormous majority of genetic profiles.

## Figures and Tables

**Figure 1 microorganisms-11-01472-f001:**
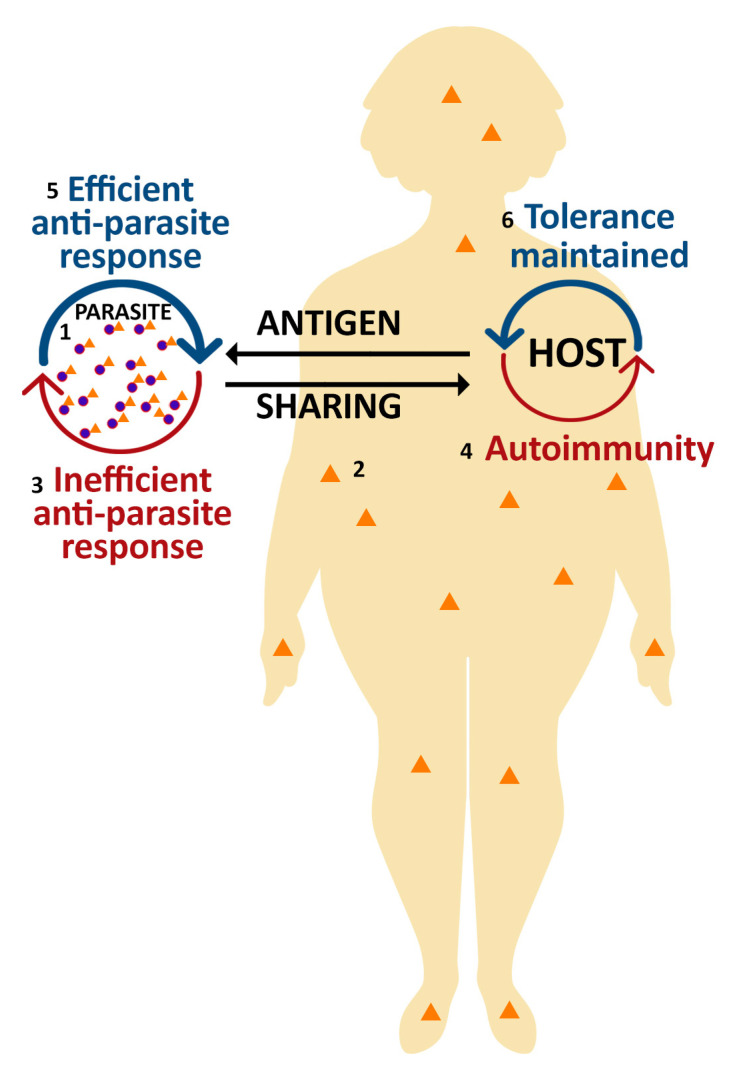
The concept of “molecular mimicry” considers that an infecting organism (parasite, bacteria, and viruses) (**1**) that shares antigens (orange triangles) (**2**) with its host has increased chances of survival by escaping the immune response triggered by the microorganism itself (**3**). The price of this evolutionary advantage, beneficial for the parasite, is the risk that this sharing of antigens generates a host immune response to its own constituents, that is; an autoimmune response (**4**). Notwithstanding, the more common responses, represented by the ticker blue half-arc arrows, are the efficient anti-parasitic response (**5**) and the maintenance of immune tolerance (**6**).

**Figure 2 microorganisms-11-01472-f002:**
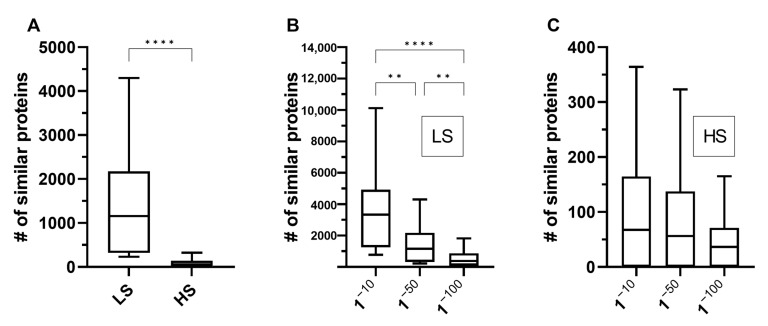
Similar proteins between humans and selected organisms. The number of similar proteins to the human proteome varied significantly when low stringency parameters (sequence coverage greater than 30% and identity greater than 10%) were changed to high (sequence coverage greater than 70% and identity greater than 70%) and as e-values changed from 1^−10^ to 1^−100^ (a complete list of all compared organisms is provided in Appendix A). (**A**) Number of similar proteins with low stringency (LS) and high stringency (HS) parameters when considered an e-value of 1^−50^. (**B**) Number of similar proteins with LS parameters (minimum percentage coverage of 30% and minimum percentage identity of 10%). (**C**) Number of similar proteins with HS parameters (minimum percentage coverage of 70% and minimum percentage identity of 70%). The data were analyzed using a Mann–Whitney test (**A**) or a Kruskal–Wallis test followed by Dunn’s multiple comparison post-tests (**B**,**C**). **, *p* < 0.005; ****, *p* < 0.0001.

**Figure 3 microorganisms-11-01472-f003:**
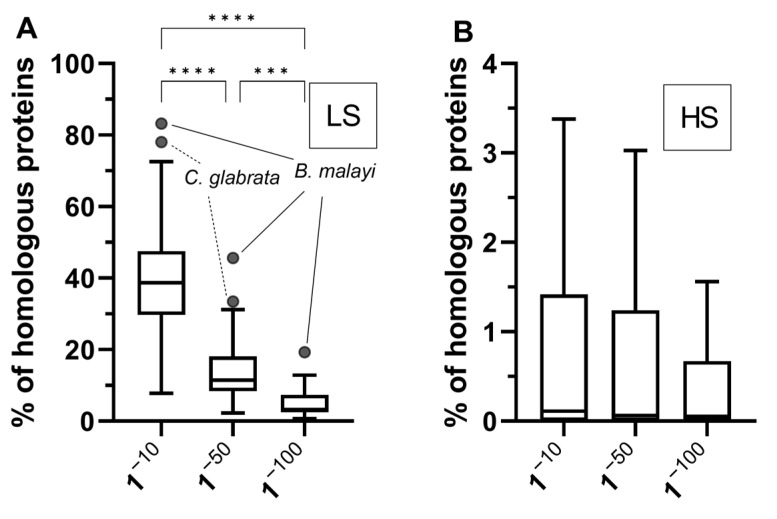
Percentage of similar proteins between humans and common microorganisms. The percentage of similar proteins to the human proteome was calculated from the count of similar hits in relation to the total number of proteins of a given species. (**A**) Proportion of homologous proteins with minimum percentage low stringency (LS) parameters (coverage of 30% and minimum percentage identity of 10%). Ratios varied significantly as e-values changed from 1^−10^ to 1^−100^. (**B**) Proportion of homologous proteins with high stringency (HS) parameters (minimum percentage coverage of 70% and minimum percentage identity of 70%). The data were analyzed using a Kruskal–Wallis test followed by Dunn’s multiple comparison post-tests. ***, *p* < 0.001; ****, *p* < 0.0001.

**Figure 4 microorganisms-11-01472-f004:**
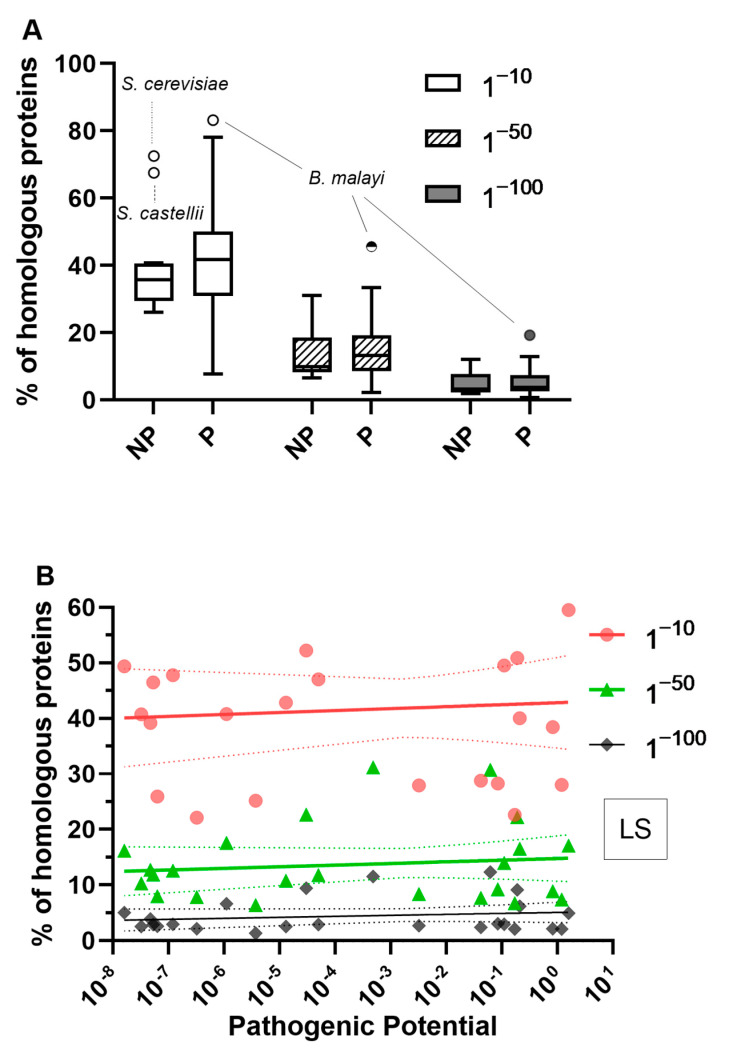
The degree of antigen sharing is not related to pathogenicity or virulence. (**A**) Percentage of similar proteins between humans and common pathogenic and non-pathogenic organisms. The percentage of similar proteins to the human proteome was calculated from the count of similar hits in relation to the total number of proteins of a given species. Proteins were considered homologous when presenting minimum percentage coverage of 30% and minimum percentage identity of 10%. Ratios varied significantly in the same group as e-values changed from 1^−10^ to 1^−100^. The data were analyzed using a Kruskal–Wallis test followed by Dunn’s multiple comparison post-tests (**B**,**C**). Correlation between the pathogenic potential of 24 species and their percentage of similar proteins to the mice proteome. Low stringency parameters (LS, sequence coverage greater than 30% and identity greater than 10%, (**B**) and high stringency parameters (HS, sequence coverage greater than 70% and identity greater than 70%, (**C**) were analyzed. Spearman correlation coefficients and two-tailed *p* values were calculated. For better visualization of the data, nonlinear regression was done to interpolate a semi-log solid-line and 90% CI bands (dotted lines) for each e-value analyzed. NP = non-pathogenic, P = pathogenic.

## Data Availability

The data presented in this study are openly available in Dryad at https://doi.org/10.5061/dryad.5dv41ns91 (accessed on 4 May 2023), reference number 0000-0003-4220-4979.

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
