# Peer review of "Visiting Molecular Mimicry Once More: Pathogenicity, Virulence, and Autoimmunity"

_microorganisms, 2023, doi:10.3390/microorganisms11061472_

Round 1

Reviewer 1 Report

The manuscript entitled "Visiting molecular mimicry once more: pathogenicity, virulence, and autoimmunity." Title, abstract and overall rationale of work is well written. However, there are still some major concerns, which needs to be addressed before publication.

1) Author need update the data in line number 47-49 (Although steadily risen throughout Westernized societies, it is estimated that the prevalence of autoimmune diseases is limited to around 3-10% of the world population) because they talking about the autoimmune diseases and they put references 2012.

2) Figure 1 author need explain details and this picture showing half information. I recommend the author they should revise and represent well manner.

3) In this section (Molecular mimicry and its implication as a mechanism of parasite escape from the host immune response) author must be describe details mechanism and incorporate one figure to show clear picture.

Do not repeat many times about this sentence (Considering that the maintenance of tolerance to self-antigens is the rule, Raymond Damian coined the concept of “molecular mimicry”, in 1964, based on the observation of antigen sharing between parasites and hosts).

4) Figure 2 and 3 need to increase font size and resolution

5) Most of the references are old and author need to revise.

Author Response

Response to Reviewer 1 Comments

The manuscript entitled "Visiting molecular mimicry once more: pathogenicity, virulence, and autoimmunity." Title, abstract and overall rationale of work is well written. However, there are still some major concerns, which needs to be addressed before publication.

1) Author need update the data in line number 47-49 (Although steadily risen throughout Westernized societies, it is estimated that the prevalence of autoimmune diseases is limited to around 3-10% of the world population) because they talking about the autoimmune diseases and they put references 2012.

A: We changed the 2 references previously provided to a more recent one. New reference 6 (Miller FW. The increasing prevalence of autoimmunity and autoimmune diseases: an urgent call to action for improved understanding, diagnosis, treatment, and prevention. Curr Opin Immunol. 2023;80:102266.)

2) Figure 1 author need explain details and this picture showing half information. I recommend the author they should revise and represent well manner.

A: We thank this referee for this suggestion. Figure 1 was conceived and designed to illustrate the concept of “Molecular mimicry” and not to explain the mechanisms underlying its functioning, that is not in the scope of the manuscript. Therefore, we ask for the agreement and permission of this referee to maintain the figure as it is.

3) In this section (Molecular mimicry and its implication as a mechanism of parasite escape from the host immune response) author must be describe details mechanism and incorporate one figure to show clear picture.

A: Thank you for your suggestion. We introduced, in this section of the manuscript, a sentence to better explain the mechanisms by which the phenomenon of molecular mimicry would operate generating tolerance to antigens shared between hosts and parasites (lines 126-126). We also pointed to a recent review that goes into the details of host tolerance mechanisms associated with molecular mimicry (Reference number 3. Rojas, M.; Restrepo-Jimenez, P.; Monsalve, D.M.; Pacheco, Y.; Acosta-Ampudia, Y.; Ramirez-Santana, C.; Leung, P.S.C.; Ansari, A.A.; Gershwin, M.E.; Anaya, J.M. Molecular mimicry and autoimmunity. J Autoimmun 2018, 95, 100-123, doi:10.1016/j.jaut.2018.10.012.). However, we believe that describing detailed mechanisms by which parasites use molecular mimicry to scape host immune response is beyond the scope of the manuscript.

Do not repeat many times about this sentence (Considering that the maintenance of tolerance to self-antigens is the rule, Raymond Damian coined the concept of “molecular mimicry”, in 1964, based on the observation of antigen sharing between parasites and hosts).

A: We suppressed the sentence from the abstract and on the legend of the figure 1, as suggested, and the phrase appears now only once in the manuscript (lines 103 – 105).

4) Figure 2 and 3 need to increase font size and resolution

A: We provide high resolution figures attached to the manuscript. Please see attachment.

5) Most of the references are old and author need to revise.

A: We agree with the reviewer that the paper has references from the 1960s-1980s. These references are used as primary sources when we discuss the history and evolution of the concept of molecular mimicry. Therefore, revising these references would make the manuscript less scientifically rigorous. Therefore, we respectfully ask the reviewer to allow citation of these seminal Papers. The paper quotes, however, a significant number of recent references that concern questions of proteome and proteome comparison that represent the main subject and methodology approached. We can obviously review the specific reference(s) that this referee considers to be outdated and that need(s) mandatorily to be updated.

Reviewer 2 Report

The research Visiting molecular mimicry once more: pathogenicity, virulence, and autoimmunity by Yuri Chaves Martins et al. is very interesting and gives new information.

Editoral comments:

Line 64 should be (parasite, bacteria and viruses)

Line 73 should be Streptococcus pyogenes,

Line 85 should be Treponema spp. infections

Line 86 should be malaria [21].

Line 136 should be It 

Author Response

Response to Reviewer 2 Comments

 Editorial comments:

Line 65 should be (parasite, bacteria and viruses)

A: We changed as requested

Line 74 should be Streptococcus pyogenes,

A: We changed as requested.

Line 87 should be Treponema spp. Infections 

A: We changed as requested.

Line 88 should be malaria [21].

A: We changed as requested.

Line 139 (170) should be It 

A: We didn’t understand the reason.

Round 2

Reviewer 1 Report

The authors have addressed all the concerns raised in the previous version of the manuscript and the quality has much improved after incorporating required modifications. Therefore, the manuscript may be considered for publication in this Journal.

Author Response

We thank the reviewer for their comments.